# Health-Promoting Effects of Dietary Polyamines

**DOI:** 10.3390/medsci9010008

**Published:** 2021-02-05

**Authors:** Rika Hirano, Hideto Shirasawa, Shin Kurihara

**Affiliations:** 1Research Institute for Bioresources and Biotechnology, Ishikawa Prefectural University, Nonoichi, Ishikawa 921-8836, Japan; hirano@waka.kindai.ac.jp; 2Faculty of Biology-Oriented Science and Technology, Kindai University, Kinokawa, Wakayama 649-6493, Japan; 1718320003a@waka.kindai.ac.jp

**Keywords:** polyamines, food, health-promoting effect, intestinal bacteria

## Abstract

The purpose of this paper is to summarize the latest information on the various aspects of polyamines and their health benefits. In recent years, attempts to treat cancer by reducing elevated polyamines levels in cancer cells have been made, with some advancing to clinical trials. However, it has been reported since 2009 that polyamines extend the healthy life span of animals by inducing autophagy, protecting the kidneys and liver, improving cognitive function, and inhibiting the progression of heart diseases. As such, there is conflicting information regarding the relationship between polyamines and health. However, attempts to treat cancer by decreasing intracellular polyamines levels are a coping strategy to suppress the proliferation-promoting effects of polyamines, and a consensus is being reached that polyamine intake does not induce cancer in healthy individuals. To provide further scientific evidence for the health-promoting effects of polyamines, large-scale clinical studies involving multiple groups are expected in the future. It is also important to promote basic research on polyamine intake in animals, including elucidation of the polyamine balance between food, intestinal bacteria, and biosynthesis.

## 1. Introduction

Polyamines are a general term for aliphatic hydrocarbons with amino groups at both ends of its molecular structure, the major ones being putrescine, spermidine, and spermine. Polyamines, which are found in almost all living organisms, are required for normal cell growth and differentiation in prokaryotes [1,2,3] and are indispensable for the maintenance of life in eukaryotes [4]

A critical function of spermidine in eukaryotes is the activation of eukaryotic translation initiation factor 5A (eIF5A) [5]. eIF5A is an essential protein for eukaryotic protein synthesis, and its activation requires post-translational modifications. During the post-translational modification process, a specific eIF5A lysine residue is changed to a hypusine residue by sequential reactions catalyzed by deoxyhypusine synthase, where spermidine is transfer to the lysine residue, and deoxyhypusine hydroxylase. In other words, spermidine is deeply involved in protein synthesis because it acts as a substrate for the maturation of eIF5A.

Polyamines are positively charged under physiological pH conditions and weakly bind to negatively charged intracellular molecules, such as nucleic acids, phospholipids, and ATP. Among these, polyamines bind most frequently to RNA and regulate protein translation by influencing the structure of mRNA [6]. There are 17 proteins in *E. coli* and six proteins in eukaryotes that have been experimentally reported to be translationally controlled in this manner, and they are considered to be members of the “polyamine modulon” [7]. These proteins are involved in cell proliferation, biofilm formation, enhancement of cell activity, and detoxification. Thus, polyamines are essential for the maintenance of normal activities of organisms.

There are three sources of polyamines: oral intake [8], intestinal microbiota [9,10], and biosynthesis in human cells (Figure 1). It is known that polyamine biosynthetic ability decreases with age [8], and it is currently difficult to regulate. Therefore, in order to control the polyamine concentration in the body, it is necessary to control the amount of polyamines derived from food and intestinal bacteria. As such, many studies have focused on this issue.

Since polyamines are essential for cell proliferation, they are found in high concentrations in actively proliferating cells [11], with bacterial cells in the logarithmic growth phase [10] and cancer cells [12] being typical examples. At the individual level, the concentration of polyamines in tissues is high at a young age and decreases with aging [8]. In recent years, attempts to treat cancer by reducing elevated polyamines levels in cancer cells have been made, with some advancing to clinical trials [13]. However, it has been reported since 2009 that polyamines extend the healthy life span of animals by inducing autophagy, protecting the kidneys and liver, improving cognitive function, and inhibiting the progression of heart diseases (Table 1). Moreover, clinical trials have been initiated in Europe in response to these findings.

The trend in Europe regarding polyamines as a health food is developing rapidly, and the global market for polyamines is expected to expand greatly if scientific evidence is sufficiently provided through future clinical trials. The purpose of this paper is to summarize the latest information regarding the various aspects of polyamines and their health benefits.

## 2. Health Benefits of Oral Ingestion of Polyamines

Soda et al. first reported in 2009 about the effects of polyamines on extending healthy lifespans in mice [14] (Table 1). In that report, the concentration of polyamines in the blood of 50-week-old mice fed with high polyamine chow containing spermidine and spermine at 374 nmol/g (54 μg/g) and 1540 nmol/g (310 μg/g), respectively, from 24 weeks of age was higher when compared to that of mice fed with low-polyamine chow containing spermidine and spermine at 143 nmol/g (21 μg/g) and 224 nmol/g (45 μg/g), respectively, from 24 weeks of age. At 88 weeks of age, the incidence of glomerulosclerosis was lower, and the expression of the senescence marker protein-30 (SMP30), which decreases with aging, was shown to be increased in the kidneys and liver of mice fed with high-polyamine chow when compared to that in mice fed with low-polyamine chow. Furthermore, the survival rate of mice fed with high-polyamine chow was significantly higher than that of mice fed with low-polyamine chow. Therefore, the results of the study showed that oral intake of polyamines can improve overall health and promote healthy life expectancy.

Around the same time as Soda’s report, Madeo et al. reported that longevity was greatly extended in yeast, flies, worms, and human cells treated with spermidine [15]. In this study, spermidine administration inhibited oxidative stress in aging mice and deacetylated histone H3 through the inhibition of histone acetyltransferases in aging yeast. Additionally, the altered acetylation status of chromatin was shown to upregulate the expression of autophagy-related genes and promoted autophagy in yeast, flies, worms, and human cells. Furthermore, Soda et al. reported that the lifespan of mice fed with high-polyamine chow was significantly longer than that of mice fed with low-polyamine chow, which was due to the suppression of abnormal gene methylation via an increase in DNA methyltransferase activity [16].

In 2013, Sigrist et al. reported that dietary spermidine supplementation in aging flies suppressed age-induced memory impairment via autophagy mechanisms. Moreover, they showed that ornithine decarboxylase-1, the rate-limiting enzyme for polyamine synthesis, suppressed olfactory memory loss in aged flies when expressed specifically in Kenyon cells, which are crucial for olfactory memory formation [18]. In addition, in 2016, Madeo et al. reported that oral supplementation with spermidine in mice extended the lifespan, reduced cardiac hypertrophy and systemic blood pressure, enhanced cardiac autophagy, and improved the mechano-elastic properties of cardiomyocytes [20].

As described above, research on the health-promoting effects of polyamines has progressed rapidly in the 21st century and the development pace is increasing, with heightened interest in their effects on the extension of healthy longevity.

## 3. Foods Containing High Concentrations of Polyamines

There have been many reports on polyamine concentrations in various foods [8,21,22,23,24,25,26,27,28,29,30,31,32,33,34,35,36,37,38,39,40,41], with a review being published in 2019 [42]. Polyamines are found in all types of food and in a wide range of concentrations. Spermidine and spermine are naturally present in many foods. The main polyamine in plant-derived foods is spermidine, whereas spermine content is generally higher in animal-derived foods. In addition, the concentration of polyamines in fermented foods is high, and microorganisms are considered to play a major role in the accumulation of polyamines in foods.

A 2006 study that measured the polyamine content of 227 foods [8] showed that turban shell viscera was the food richest in polyamines, with the total amount of putrescine, spermidine, and spermine being 124 μmol/g (20,000 μg/g). However, there were large individual differences, which are possibly due to differences in the microbiota of the turban shells. The second food richest in polyamines was wheat germ, with 3.86 μmol/g (560 μg/g) of polyamines, followed by dried agaricus mushrooms (400 μg/g) in third place, green peppers (390 μg/g) in fourth place, and chicken liver (340 μg/g) in fifth place.

The estimated dietary intake of polyamines is reported to be 42 mg/day [43] in Europe (UK, Italy, Spain, Finland, Sweden, and the Netherlands), 29 mg/day [44] in the US,16 mg/day [45] in Turkey, 26 mg/day in Japan [34], and 36 mg/day in Sweden [46].

In Japan, there are several supplements on the market that are designed to provide an efficient intake of polyamines. "Soypolia" (polyamine content: 1.5 mg/g or more, according to the manufacturer’s website), which is made from soybean extracts and produced by Combi, and “Oryzapolyamine” (polyamine content: 2 mg/g or more, according to the manufacturer’s website), which is made from rice embryo extracts and produced by Oryza Oil & Fat Chemical.

## 4. Health Benefits of Polyamines Derived from Gut Microbiota

A promising source of polyamines other than food is the commensal bacteria in the gut. Although the concentration of polyamines in feces varies greatly among individuals, the concentrations are a maximum of several mM for putrescine (several hundred μg/g) and a maximum of several hundred μM for spermidine (several hundred μg/g) [47], which is comparable to those found in the polyamine-rich foods described below. In 2011, Matsumoto reported that the administration of *Bifidobacterium animalis* subsp. *lactis* LKM512 in mice increased polyamines levels in the intestine and significantly extended the lifespan, which was associated with the downregulation of inflammation-associated genes and improvement of the intestinal barrier function [17]. In 2014, the same group reported that orally administered arginine, a precursor of polyamines, enhanced polyamine production by intestinal bacteria, inhibited systemic inflammation, promoted memory, and prolonged longevity in mice [19].

The colon has traditionally been thought of as an organ that absorbs water and metal ions. However, recent studies on gut bacteria have reported that acetate produced by bifidobacteria improves the defense functions of epithelial cells [48]. Moreover, the production of indole propionic acid, which has neuroprotective properties [49], by gut commensal bacteria is necessary to maintain indole propionic acid levels in serum [50]. Thus, the metabolites produced by gut microbes are absorbed into the body through the colon epithelium and have a large impact on the health of the host. Although most of the polyamines in the intestinal lumen are absorbed in the duodenum and ileum [51], it has been suggested that polyamines produced by the conversion of arginine by intestinal microbiota in the colon enter the bloodstream [19]. Another report suggests that polyamines in the colon enter the portal circulation, as [^14^C] putrescine instilled into the lumen of the gut was detected in the pancreaticobiliary secretion [52]. The concentration of polyamines in the colonic lumen is thought to be determined by the equilibrium between polyamine uptake (utilization) and export (production) by intestinal bacteria, and many of the most predominant species of indigenous human intestinal microbiota have been reported to take up and/or export polyamines [10]. The development of methods to regulate the levels of polyamines in the colonic lumen, such as the aforementioned increase in the concentration of polyamines by simultaneous administration of bifidobacteria and arginine [19], is in progress, and a portion of the methods has already been commercialized. For example, LKM512 bifidobacteria, a probiotic that promotes the production of polyamines by gut microbiota [17,53], is also commercialized and is sold by Kyodo Milk Industry.

## 5. Clinical Trials Regarding Polyamines-Induced Health Improvements

In Europe, clinical trials focusing on the ability of polyamines to improve cognitive function began in 2018 [54], and prior to this, the safety and tolerability of wheat germ-derived spermidine used in oral polyamine consumption clinical trials was confirmed [55]. In a randomized, double-blind, placebo-controlled phase IIa trial reported in 2018, older adults aged 60–80 years (n = 15 in the polyamine and control groups) received 1200 mg spermidine per day for 3 months, with the polyamine group reporting statistically significant improvements in cognition when compared to the control group [54]. Based on the results [11], a randomized, double-blind, placebo-controlled phase IIb trial, which began in 2019, will analyze the effects of concurrently administering 200 μg of putrescine, 900 μg of spermidine, and 500 μg of spermine (1600 μg total) per day for 12 months in older adults aged 60 to 90 years (n = 50 in the polyamine and control groups). The amount of polyamines administered in this clinical trial is approximately 4–10% of the estimated dietary intake of polyamines in developed countries [43,44]. On the other hand, in animal experiments in which the various health-promoting effects of polyamines have been reported, the polyamine intake in the high-polyamine group was several times higher than that in the control group. Therefore, it would be interesting to see the results of administering small amounts of polyamines in humans. Since the no observable adverse effect level is 13.5 μg/kg body weight per day for spermidine or 3.1 μg/kg body weight per day [56] for spermine, the amount of polyamine administered in this clinical trial is only approximately 0.1% of the no adverse effect dose for an adult weighing 60 kg [11,54], and therefore, their safety is considered to be sufficiently ensured.

## 6. Discussion

The fact that there is conflicting information regarding the relationship between polyamines and health might be confusing. On the one hand, decreased polyamines levels are used to treat cancer, while on the other hand, increased polyamines levels are thought to extend healthy life span. Clinical studies have shown that the simultaneous administration of Sulindac, a known anti-inflammatory agent, and D,L-α-Difluoromethylornithine (DFMO), an inhibitor of polyamine synthesis, is effective in the treatment of cancer [13]. This is thought to be due to the inhibition of the proliferation-promoting effect of polyamines on cancer cells. However, polyamine intake does not seem to induce carcinogenesis in healthy individuals. For example, a 2015 epidemiological study of 87,602 postmenopausal women on polyamine intake and the risk of developing colorectal cancer found no positive association between dietary polyamines and colorectal cancer [57].

The health-promoting effects of polyamine intake, which have been reported since the beginning of the 21st century (Table 1), are attracting worldwide attention. Thus, large-scale clinical studies by multiple groups are expected in the future. In ongoing clinical studies in Europe [11], oral polyamine is administered at concentrations that account for only 4–10% of the estimated average dietary intake. One reason for clinical studies in humans beings limited to small doses of polyamines is that the polyamine concentration in the wheat germ-derived supplements used is not high enough to provide large polyamine concentrations in the administered volume. Therefore, in order to extend the healthy life span of humans via polyamine supplementation, it is important to develop materials that contain high concentrations of polyamines. Polyamines in polyamine-rich foods such as turban shell, eel liver, and pond smelt [8] can be derived from the intestinal microbiota of the respective aquatic organisms. In addition, polyamines, which are often found in high concentrations in pickled vegetables, such as Shibazuke pickles [8], could be derived from food-fermenting lactic acid bacteria. If dietary polyamines can be produced by fermentation methods via the isolation of high polyamine-producing bacteria from these food-experienced bacteria, it will be possible to develop products that contain high concentrations of polyamines (Figure 2).

It would also be effective to develop technologies that enhance polyamine production by intestinal indigenous bacteria, which are an important source of polyamines. There are constantly 100–200 g of feces in the large intestine, which contain several hundred μg of polyamines per g of feces [47] derived from intestinal indigenous microorganisms [9]. It is also important to promote basic research on polyamine intake in animals, including the elucidation of the balance of polyamine intake between food, intestinal bacteria, and biosynthesis.

## Figures and Tables

**Figure 1 medsci-09-00008-f001:**
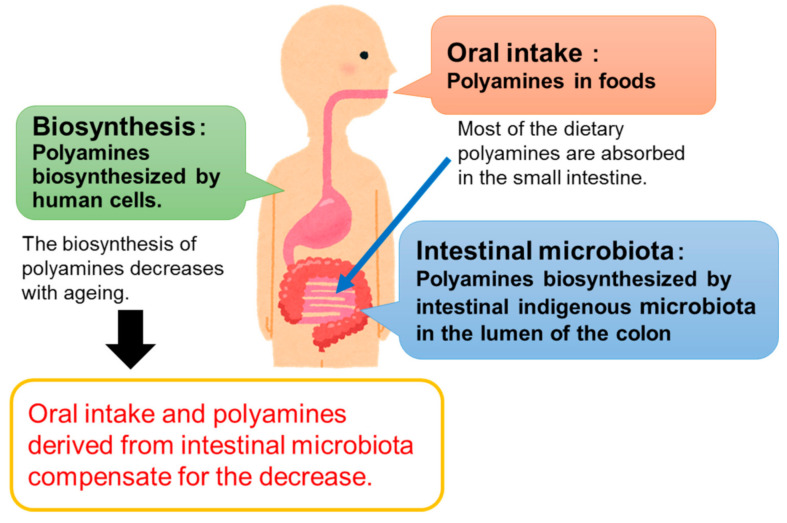
Sources of polyamines in animals.

**Figure 2 medsci-09-00008-f002:**
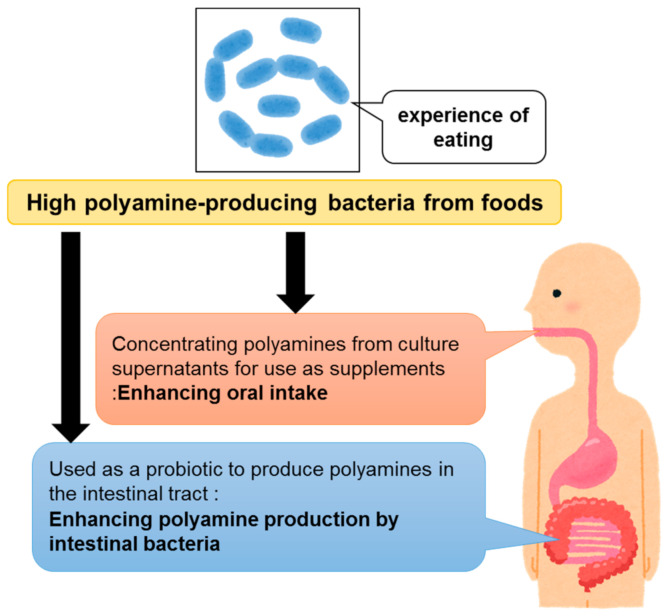
Significance of isolating high polyamine-producing bacteria from foods.

**Table 1 medsci-09-00008-t001:** Health promotion and life extension by polyamines.

Effects	Experimental Animals	Mechanisms	Polyamine Sources	Literature
Life extension	Mice	Protection of the kidneys and liver	Oral ingestion	[14]
Life extension	Mice and Flies	Autophagy induction	Oral ingestion	[15]
Life extension	Mice	Inflammation suppression	Gut microbiota	[17]
Memory enhancement	Flies	Autophagy induction	Oral ingestion	[18]
Life extension	Mice	Inhibition of abnormal methylation	Oral ingestion	[16]
Life extension and improved cognition	Mice	Inflammation suppression	Gut microbiota	[19]
Improved heart function	Mice	Autophagy induction	Oral ingestion	[20]

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
