# Peer review of "Health-Promoting Effects of Dietary Polyamines"

_medsci, 2021, doi:10.3390/medsci9010008_

Round 1

Reviewer 1 Report

The review by Hirano et al. summarizes the main findings related to the beneficial effects of dietary polyamines. I have the following comments:

1/ Line 29: replace the word “polyamines” by “spermidine”.

2/ Lines 29-36: In this paragraph, the authors need to introduce the enzyme DHPS and DOHH.

3/ Lines 35-36: I am not agree with this statement. It has to be justified by publications or remove.

4/ The paragraph lines 46-51 is not clear. It needs revision. Remove the word “for” line 46 and revise the use of “regulate”.

5/ Lines 53-55: Include refs to support these statements, notably regarding cells in contact with growing bacteria.

6/ Include space between words and refs on lines 56 and 58.

7/ After the introduction, I would suggest to reorganize the review. To my opinion, the authors should follow this plan: 2/ Sources of polyamines (foods containing high concentrations of polyamines and the role of the microbiota); 3/ Experimental evidence of the benefits of polyamine consumption (all the result obtained in animals); 4/ Human data (including the works by Madeo’s group and the clinical trials). These titles are just giving as examples.

8/ The authors need to indicate that polyamines are absorbed in the small intestine. The sentence lines 115-117 brings the reader to think that maybe polyamines in the colon lumen can be absorbed by CECs. Again, either the authors use a reference to support this argument or they remove it.

9/ The paragraph about the role of the microbiota on polyamine synthesis needs a conclusion. In addition, the authors should discuss about the use of polyamines by the microbiota.

10/ Line 176: be more specific about which cancers are treated by DFMO alone. For example, to my recollection, it has been never described that DFMO alone reduces colorectal cancer.

11/ The following paper has to be cited: Vargas AJ et al. Dietary polyamine intake and colorectal cancer risk in postmenopausal women. Am J Clin Nutr 2015;102:411-419.

Author Response

Thank you very much for the peer review. We have revised the manuscript according to the reviewer's suggestions.

Point 1: Line 29: replace the word “polyamines” by “spermidine”.

Response 1: In accordance with the reviewer's suggestion, we replaced the word “polyamines” by “spermidine” in Line 29.

Point 2: Lines 29-36: In this paragraph, the authors need to introduce the enzyme DHPS and DOHH.

Response 2: In accordance with the reviewer's suggestion, we introduced deoxyhypusine synthase and deoxyhypusine hydroxylase in Lines 33-34.

Point 3: Lines 35-36: I am not agree with this statement. It has to be justified by publications or remove.

Response 3: In accordance with the reviewer's suggestion, we deleted the sentence “Moreover, it is thought to be the reason why the loss of genes in the polyamine synthesis system is lethal in eukaryotes.” in Line 36.

Point 4: The paragraph lines 46-51 is not clear. It needs revision. Remove the word “for” line 46 and revise the use of “regulate”.

Response 4: In accordance with the reviewer's suggestion, we revised the sentence in Lines 48-49.

Point 5: Lines 53-55: Include refs to support these statements, notably regarding cells in contact with growing bacteria.

Response 5: In accordance with the reviewer's suggestion, we included refs 10-12 in Lines 54-55.

Point 6: Include space between words and refs on lines 56 and 58.

Response 6: In accordance with the reviewer's suggestion, we added spaces in Line 56 and 58.

Point 7: After the introduction, I would suggest to reorganize the review. To my opinion, the authors should follow this plan: 2/ Sources of polyamines (foods containing high concentrations of polyamines and the role of the microbiota); 3/ Experimental evidence of the benefits of polyamine consumption (all the result obtained in animals); 4/ Human data (including the works by Madeo’s group and the clinical trials). These titles are just giving as examples.

Response 7: In accordance with the reviewer's suggestion, we reorganized the manuscript. However, for the purpose of attracting the reader's attention, the health benefits of polyamines were placed right after the introduction. For the rest, we followed the reviewer's suggestion.

Point 8: The authors need to indicate that polyamines are absorbed in the small intestine. The sentence lines 115-117 brings the reader to think that maybe polyamines in the colon lumen can be absorbed by CECs. Again, either the authors use a reference to support this argument or they remove it.

Response 8: In accordance with the reviewer's suggestion, we added references 19, 51 and 52 to support the argument in Lines 151-156.

Point 9: The paragraph about the role of the microbiota on polyamine synthesis needs a conclusion. In addition, the authors should discuss about the use of polyamines by the microbiota.

Response 9: In accordance with the reviewer's suggestion, we discussed about the use of polyamines by the microbiota in Lines 156-164.

Point 10: Line 176: be more specific about which cancers are treated by DFMO alone. For example, to my recollection, it has been never described that DFMO alone reduces colorectal cancer.

Response 10: In accordance with the reviewer's suggestion, we revised the manuscript in Lines 197-201.

Point 11: The following paper has to be cited: Vargas AJ et al. Dietary polyamine intake and colorectal cancer risk in postmenopausal women. Am J Clin Nutr 2015;102:411-419.

Response 11: In accordance with the reviewer's suggestion, we added the reference and the sentences in Lines 201-205.

Reviewer 2 Report

The authors, Rika Hirano, Hideto Shirasawa and Shin Kurihara presented here their article on the study of “Health-promoting effects of dietary polyamines”.

The closest work to this review article is “Polyamines in Food” published in 2019 (Front Nutr. 2019; 6: 108). However, the current work put more emphasis on the efficacy of dietary polyamines, summarized the effects observed in polyamines fed animals, the results of recent human clinical trials, and foods rich in polyamines. I think the content of this article can attract readers' interest. The logic of the article is clear and the sentences are smooth. But I think that the section of “Foods containing high concentrations of polyamines” can include more contents. The abovementioned previous work should be cited and make some comparisons and comments of it. It will increase the readability of this paper.

Author Response

Point 1: The closest work to this review article is “Polyamines in Food” published in 2019 (Front Nutr. 2019; 6: 108). However, the current work put more emphasis on the efficacy of dietary polyamines, summarized the effects observed in polyamines fed animals, the results of recent human clinical trials, and foods rich in polyamines. I think the content of this article can attract readers' interest. The logic of the article is clear and the sentences are smooth. But I think that the section of “Foods containing high concentrations of polyamines” can include more contents. The abovementioned previous work should be cited and make some comparisons and comments of it. It will increase the readability of this paper.

Response 1: Thank you very much for giving our paper a favourable evaluation. In accordance with the reviewer's opinion, we cited Front Nutr. 2019; 6: 108 and added description of polyamine distribution in foods in Lines 108-114.

Round 2

Reviewer 1 Report

The authors responded to all my critics.